# Microstructure, Mechanical Property and Thermal Conductivity of Porous TiCO Ceramic Fabricated by In Situ Carbothermal Reduction of Phenolic Resin and Titania

**DOI:** 10.3390/nano14060515

**Published:** 2024-03-13

**Authors:** Xiaoyu Cao, Chenhuan Wang, Yisheng Li, Zehua Zhang, Lei Feng

**Affiliations:** Shaanxi Key Laboratory of Green Preparation and Functionalization for Inorganic Materials, School of Material Science and Engineering, Shaanxi University of Science and Technology, Xi’an 710021, China; 210211012@sust.edu.cn (C.W.); 220211035@sust.edu.cn (Y.L.); 230211041@sust.edu.cn (Z.Z.); fenglei@sust.edu.cn (L.F.)

**Keywords:** porous ceramic, carbothermal reduction, catalyst content, mechanical property, thermal conductivity

## Abstract

The porous TiCO ceramic was synthesized through a one-step sintering method, utilizing phenolic resin, TiO_2_ powder, and KCl foaming agent as raw materials. Ni(NO_3_)_2_·6H_2_O was incorporated as a catalyst to facilitate the carbothermal reaction between the pyrolytic carbon and TiO_2_ powder. The influence of Ni(NO_3_)_2_·6H_2_O catalyst content (0, 5, 10 wt.% of the TiO_2_ powder) on the microstructure, compressive strength, and thermal conductivity of the resultant porous TiCO ceramic was examined. X-ray diffraction and X-ray photoelectron spectroscopy results confirmed the formation of TiC and TiO in all samples, with an increase in the peak of TiC and a decrease in that of TiO as the Ni(NO_3_)_2_·6H_2_O content increased from 0% to 10%. Scanning electron microscopy results demonstrated a morphological change in the pore wall, transforming from a honeycomb-like porous structure composed of well-dispersed carbon and TiC-TiO particles to rod-shaped TiC whiskers, interconnected with each other as the catalyst content increased from 0% to 10%. Mercury intrusion porosimetry results proved a dual modal pore-size distribution of the samples, comprising nano-scale pores and micro-scale pores. The micro-scale pore size of the samples minorly changed, while the nano-scale pore size escalated from 52 nm to 138 nm as the catalyst content increased from 0 to 10%. The morphology of the pore wall and nano-scale pore size primarily influenced the compressive strength and thermal conductivity of the samples by affecting the load-bearing capability and solid heat-transfer conduction path, respectively.

## 1. Introduction

The escalating flight speed of the hypersonic aircraft has resulted in increasingly severe thermal conditions due to aerodynamic heating. These conditions necessitate superior high-temperature stability, mechanical performance, and thermal insulation properties of the surface thermal protection materials [1]. Porous ceramics are frequently examined as thermal insulation materials due to their distinctive features, such as high porosity, low bulk density, low thermal conductivity, and exceptional resistance to high temperatures and corrosion [2]. Porous ceramics, such as SiO_2_ [3], Al_2_O_3_ [4], ZrO_2_ [5], SiC [6,7,8,9,10], TiC [11,12], ZrC [13,14,15], and HfC [16,17], exhibit outstanding thermal insulation performances characterized by high melting points, excellent high-temperature stability, and robust mechanical properties, rendering them highly promising for high-temperature insulation applications.

At present, several methods exist for the fabrication of porous ceramics, including sintering [15,16,18,19,20,21], sacrificial templating [22,23,24,25], sol-gel [26,27,28,29], and polymer precursor conversion (PDC) [30,31] techniques. The sintering technique primarily employs ceramic particles and pore-forming agents as raw materials, producing porous carbide ceramics through high-temperature sintering around 2000 °C, necessitating a sophisticated experimental apparatus and consequently, a high cost. For instance, Chen et al. [16] successfully prepared porous ZrC and HfC using a sintering method and achieved uniform pore structures with porosities of 68.74% and 77.82%, room temperature thermal conductivities of 1.12 and 1.01 W/(m·K), and compressive strengths of 8.28 and 5.51 MPa, respectively. Shao et al. [21] used pressureless-reaction sintering combined with a foam-gel casting freeze-drying technique to prepare a novel highly porous high-entropy (Zr_1/5_Hf_1/5_Nb_1/5_Ta_1/5_Ti_1/5_) ceramic. The sacrificial templating method is another option for preparing porous carbide ceramics, but it has limitations in terms of ceramic composition and increases experimental costs due to the use of pore-forming agents. Alternatively, Biggemann et al. [25] used pyrolyzed cellulose fibers and phenolic resin spheres as pore-forming agents to prepare porous alumina ceramics. They found that the thermal conductivity and strength of the porous alumina ceramics strongly depend on the multimodal combination of sacrificial templates and the resulting pore network. The porous carbide ceramics prepared by the sol-gel method are characterized by uniform mesoporous pore size distribution and low environmental pollution [28]. However, special drying techniques, such as supercritical fluids drying, freeze drying, or exhausted solvent exchange, should be applied during sol-gel formation. Time consumption and high costs are the major drawbacks of sol-gel methods that limit their large-scale production and future applications. Liu et al. [29] prepared a porous lanthanum zirconate ceramic with a sol-gel method with compressive strengths of 7.1–7.8 MPa and a thermal conductivity of 0.033 W/(m·K), showing excellent high-temperature heat insulation performance. The PDC method facilitates the control of ceramic composition and nano-pore structure, thereby reducing thermal conductivity. The synthesis of ceramic precursors required for the PDC method is complex and challenging, leading to costly and limited precursor options. This complexity hinders the widespread adoption of the PDC method. Wang et al. [30] employed a liquid hyperbranched polycarbosilane as a precursor to fabricate porous silicon carbide ceramics. A highly porous SiC foam with a density of 0.211 g cm^−3^ and porosity of 91.2% was obtained, and its thermal conductivity at room temperature was as low as 0.05 W/(m·K). Currently, SiC and ZrC are the most extensively researched porous ceramics produced by the abovementioned method. Porous TiC ceramic is anticipated to exhibit lower thermal conductivity compared to other variants, such as SiC and ZrC, with equivalent pore size and particle structure attributed to its inherently lower thermal conductivity. Nevertheless, the exploration of porous TiC ceramic is limited, primarily due to challenges in synthesis and the scarcity of appropriate raw materials. The fabrication of porous TiC ceramic through a cost effective, efficient, and scalable production method is of paramount importance.

The synthesis of porous ceramics (e.g., TiC, SiC, HfC) through carbothermal reduction often encounters incomplete reactions due to the high melting points of these materials. Sun et al. [32] developed a molten-salt-assisted aluminum/carbothermal reduction technique to produce Al_2_O_3_-SiC powders. Their findings indicate that a NaCl-KCl molten salt medium markedly enhances the phase transition from quartz to SiC and lowers the composite powder’s synthesis temperature. Fu et al. [33] synthesized HfC nanowires on carbon fibers through the thermal pyrolysis of an Hf-containing organic precursor. The HfC nanowires were produced with carbothermal reduction, involving the reaction between pyrolyzed HfO_2_ (HfO) and the carbon source from the Hf-containing precursor, facilitated by a Ni catalyst derived from Ni(NO_3_)_2_·6H_2_O, which acted as a nucleation site for nanowire growth. This study presents an expedited one-step sintering process for the synthesis of porous TiCO ceramics. The method involves a carbothermal reduction at 1500 °C, utilizing phenolic resin as the carbon source and TiO_2_ particles as the titanium source. Ni(NO_3_)_2_·6H_2_O as a catalyst was utilized to facilitate the complete reaction of TiO_2_ with pyrolytic carbon. Varying concentrations of Ni(NO_3_)_2_·6H_2_O catalyst were employed to investigate its impact on the microstructure, mechanical property, and thermal conductivity of the resultant porous ceramic. The resultant porous ceramics exhibit a porosity of 63~82%, a compressive strength of 0.72~1.89 MPa, and a room-temperature thermal conductivity of 0.23~0.40 W/(m·K), depending on the content of Ni(NO_3_)_2_·6H_2_O catalyst. The high-temperature thermal conductivity of the porous TiCO ceramics exhibits minimal variation, indicating a robust, high-temperature stability of the thermal insulation property. This research validates the feasibility and potential of the developed synthetic strategy.

## 2. Materials and Methods

Firstly, a phenolic resin powder (prepared from a condensation polymerization reaction of phenol and formaldehyde, free phenol < 7%, gel time at 200 °C: 70–100 s, Shaanxi Taihang Impedefire Polymer Co., Ltd., Xi’an, China) was ground into powder without additional purification. The phenolic resin powder was dissolved in anhydrous ethanol. The mass ratio of phenolic resin powder and anhydrous ethanol is 5:7. Then, TiO_2_ (Beijing HWRK Chem Co., Ltd., Beijing, China), Ni(NO_3_)_2_·6H_2_O, and KCl (30 wt.% of the resin) were added in the resin solution. The carbon yield ratio of the phenolic resin was 45%, which was provided by the supplier. The mole ratio of TiO_2_ and pyrolytic carbon was set as 1:3, according to the carbothermal reduction reaction of TiO_2_ + 3C = TiC + CO. The content of the Ni(NO_3_)_2_·6H_2_O catalyst was set as 0, 5, and 10 wt.% of the TiO_2_ powder. In order to know the role of KCl and Ni(NO_3_)_2_·6H_2_O during the pore formation process, a monolithic phenolic resin and phenolic resin with KCl and Ni(NO_3_)_2_·6H_2_O (5 wt.% of TiO_2_) slurry (denoted as KCl/Ni(NO_3_)_2_·6H_2_O-5/resin) were also conducted. The mixture solution was blended using a magnetic agitator for 4 h. Secondly, the evenly mixed solution was poured into the silicone mold and sequentially dried in an oven at 80 °C for 12 h, followed by 120 °C for 3 h, 150 °C for 1 h, and finally 180 °C for 3 h. After that, three precursors of monolithic resin, KCl/Ni(NO_3_)_2_·6H_2_O/resin, and TiO_2_/KCl/Ni(NO_3_)_2_·6H_2_O/resin were obtained, respectively. Lastly, the precursors packaged by graphite paper were placed in the alumina crucible and heat treated in the tubular furnace in a flowing argon atmosphere. The furnace temperature was increased to 1500 °C and held up for 60 min. The experimental process is illustrated in Figure 1. The pyrolysis and carbothermal reduction in the precursor occurred during heat treatment. The as-received porous ceramic with a different content of Ni(NO_3_)_2_·6H_2_O catalyst was noted as TiCO-0, TiCO-5 and TiCO-10, respectively. 

The open porosity and bulk density of the samples were measured using the Archimedes method. Phase composition of the products was determined by X-ray diffraction using Cu Kα radiation (TK-XRD-201, Rigaku, Tokyo, Japan). X-ray photoelectron spectroscopy (XPS, Thermo Scientific k-Alpha, Thermo Fisher Scientific, Waltham, MA, USA) was used to further analyze the chemical composition of the samples. Pore size distributions of the porous TiCO ceramics were characterized by a mercury intrusion porosimetry (MIP, AutoPore IV 9500, Micromeritics, Norcross, GA, USA). The morphology and elemental composition were characterized by scanning electron microscopy (SEM, Mira3, Tescan, Brno, Czech Republic). Compressive strength (with dimensions of 10 × 10 × 7 mm^3^) was carried out using an electronically controlled, universal testing machine (UTM5105, Shenzhen Suns Technology, Shenzhen, China) with a crosshead speed of 0.5 mm/min. Thermal conductivity (k) was calculated according to the equation: k = ρ · α · C_p_, where ρ is the sample’s bulk density, α is its thermal diffusivity, and C_p_ is its specific heat capacity. The thermal diffusivity α (sample dimensions: 10 × 10 × 2 mm^3^) was measured using a commercial thermal conductivity testing instrument (LFA 467 HT, Netzsch, Selb, Germany). The specific heat capacity C_p_ values of the samples (with dimensions of 10 × 10 × 2 mm^3^) were measured using a laser thermal conductivity testing instrument (LFA 467, Netzsch, Selb, Germany).

## 3. Results and Discussion

### 3.1. Phase Composition

XRD patterns of the as-received porous TiCO ceramic are depicted in Figure 2. The three strong diffraction peaks at 2θ values of 36.0, 41.8, and 60.6 are present in all samples. These can be indexed to the (111), (200), and (220) reflections of cubic TiC (Fm3m). The intensity of the TiC diffraction peak is high, with a notably narrow full width at half maximum (FWHM), suggesting the formation of highly crystalline and pure TiC. Concurrently, a diffraction peak of TiO are evident across all samples, indicating a quantity of TiO_2_ is reduced to TiO at the current processing temperatures. The crystal plane indices of TiO were (100) and (101).

XPS of the samples was performed to further investigate the elemental composition. Figure 3 illustrates the peak decomposition of Titanium (Ti) elements. It is noticeable that the relative intensity of the Ti-C bond peak incrementally increases, while the Ti-O bond peak progressively decreases with an increase in the catalyst content. The trend of increase in TiC and decrease in TiO_2_ and TiO phase suggests a greater extent of the reaction between TiO_2_ and pyrolytic carbon as the catalyst content rises. When the Ni(NO_3_)_2_·6H_2_O content reaches 10%, TiO_2_ still persist in the sample. This indicates that under the current experimental conditions, TiO_2_ cannot be fully reduced to TiC. The components of the as-received porous ceramic include C, TiC, TiO_2_, and TiO; hence, it is denoted as porous TiCO ceramic. 

### 3.2. Microstructure

The morphologies of the monolithic resin and KCl/Ni(NO_3_)_2_·6H_2_O/resin precursors after being cured and sintered are shown in Figure 4. As seen from Figure 4a,c, the evaporation of anhydrous ethanol during curing leads to the formation of a porous carbon skeleton featuring pore sizes in the order of hundreds of micrometers. As seen from Figure 4b,e, KCl embeds within the phenolic resin and decomposes during the sintering process, resulting in larger pore sizes of hundreds and thousands of micrometers within the porous carbon skeleton compared to those derived from the monolithic phenolic resin. Consequently, the evaporation of anhydrous ethanol, along with the decomposition of KCl, contributes to the micro-scale pore formation within the porous carbon skeleton. Figure 4d,e illustrates that the K and Ni particles, resulting from the decomposition of KCl and Ni(NO_3_)_2_·6H_2_O, adhere to the surfaces of the pore wall. Ni(NO_3_)_2_·6H_2_O decomposes into NiO nanoparticle during the heating process. The formed NiO can be reduced to Ni particles by self-generated CO (g) derived from the carbothermal reduction process [34].

The microstructures of the received porous TiCO ceramic, with varying catalyst contents, are depicted in Figure 5. As observed from Figure 5, all samples exhibit both micro-scale and nano-scale pores. The micro-scale pores result from the volatilization of anhydrous ethanol and the decomposition of the KCl, while the nano-scale pores are formed by the gas release during carbothermal reduction. The micro-scale pore sizes of the three samples are all in the range of hundreds and thousands of micrometers. The amplification picture of the pore wall indicates its porous structure with nano-scale pores. The morphology of the pore walls strongly depend on the catalyst content. Upon examining Figure 5a,d, it can be deduced that TiC particles form due to the reaction of TiO_2_ particles and pyrolytic carbon on the surface of the pore walls in the absence of the catalyst. The pore walls of the TiCO-0 sample consist of white particles, which are evenly dispersed within a black honeycomb-like skeleton. In conjunction with the atom ratio in the EDS results, it is hypothesized that the white particles are TiO and the honeycomb-like skeleton are carbon. This suggests that a partial phase conversation from TiO_2_ to TiC occurs rather than a complete conversation in the absence of a catalyst. After adding Ni(NO_3_)_2_·6H_2_O, nickel melts into droplets at the elevated temperature and acts as nucleation sites for TiC grains, facilitating the dissolution of titanium and carbon sources into the droplets. Subsequently, this process promotes unidirectional growth of TiC whiskers and results in a greater reduction in TiO_2_ to TiC, thus decreasing the amount of carbon skeleton. This accelerative carbothermal reduction leads to the formation of a porous structure, characterized by rod-shaped TiC whiskers that interconnect both on the surface and within the pore walls. However, when the Ni(NO_3_)_2_·6H_2_O content is further increased to 10%, excessive nucleation sites lead to the agglomeration of TiC grains, which results in pore wall structures with larger pore sizes and less carbon skeleton. Based on this analysis, it can be concluded that Ni(NO_3_)_2_·6H_2_O catalyst content primarily influences the pore wall structure of the porous TiCO ceramic.

### 3.3. Pore Size Distribution

Figure 6 presents the pore-size distribution curves for samples with varying catalyst contents. The porous TiCO ceramic depicted in Figure 6 demonstrates a dual modal pore-size distribution, consisting of nano-scale pores and micro-scale pores. To clarify the effect of catalyst content on the pore structure of the porous TiCO ceramic, the average pore size of a micro-scale pore and nano-scale pore were calculated and presented in Figure 6d. As seen from Figure 6d, the average nano-scale pore size increased from 52 to 139 nm while the average micro-scale pore size minorly changed as the catalyst content increased. The nano-scale pore formation is attributed to the release of CO gas during carbothermal reduction, as mentioned previously. The extent of carbothermal reduction is influenced by the concentration of the catalyst. An increase in catalyst content leads to a higher degree of carbothermal reduction, and consequently, a larger size of nano-scale pores. These results agree with the SEM results, showing that the number and size of the nano-scale pore of the porous TiCO ceramic are obviously adjusted by changing the catalyst content.

Table 1 presents the open porosity and bulk density of the samples. The open porosity for the three samples is recorded as 63%, 75%, and 82%, respectively, while the densities are 1.05, 0.87, and 0.72 g/cm^3^, respectively. As depicted in Table 1, an increase in catalyst content generally results in an increase in open porosity and a decrease in bulk density. The increasing number and size of the nano-scale pore and the minorly changed micro-scale pore contributes to the increased open porosity of the samples as the catalyst content increases.

### 3.4. Compressive Strength

The compressive strength of the samples was measured, and the stress–strain curves are presented in Figure 7. The strain of the sample was calculated through dividing the displacement by the height of the samples. Figure 7 shows that the stress–strain curves for the three samples sharply decline after reaching their peaks, indicating a brittle fracture mode. The porous TiCO ceramic with 0, 5, and 10% catalyst content fractures at a strain of approximately 37%, 35%, and 12%. Under pressure, the cracks initiate at stress concentrations on the surface of the porous ceramic, which then propagate along the pore walls. Upon reaching the maximum load-bearing capacity of the pore walls, they collapse due to densification, resulting in sample cracking. The bearing capacity of the pore wall is directly linked to its structure, making it the key factor influencing the porous ceramic’s load-bearing ability. The pore-wall structure of the porous TiCO ceramic ranges from a honeycomb-like skeleton with an average pore size of 52 nm to interconnected rod-shaped TiC whiskers with an average pore size of 110 nm and 159 nm of the TiCO-5 and TiCO-10 samples, respectively. The honeycomb-like skeleton, having the smallest average pore size, results in the longest crack propagation path and decent elastic deformation ability, thus contributing to the highest fracture strain of the TiCO-0 sample. As the catalyst content increased, an increase in pore size leads to a shorter crack-growth path and thus a lower fracture strain. The compressive strength of the samples are shown in Table 1. As shown in Table 1, the compressive strength decreased from 1.89 to 0.072 MPa with an increase in the catalyst content. Ceramics are defect-sensitive materials, and a large pore size and uneven pore structure considerably reduces their mechanical strength. The relationship between porosity, pore size, and strength (σ) is shown in the following equation [35]:(1)$$\sigma =K\frac{\sigma_{0}\left(1-p\right)}{\sqrt{r_{m}}}$$
where *σ*_0_ is the strength at the porosity of 0, K is a constant, p is the porosity, and *r_m_* is the average pore size. The above equation means that the strength is inversely proportional to porosity and pore size. The highest compressive strength of TiCO-0 is mainly due to the lowest porosity in all the samples. The compressive strength of the samples decrease as the catalyst content increases, which is probably due to the increase in the porosity and pore size [36]. With increasing porosity, the load-bearing capacity of the porous ceramic diminishes, leading to a reduction in compressive strength. Meanwhile, the TiCO-0 sample, having the smallest average pore size, exhibits the highest pore wall strength. Increasing the catalyst content to 5% and 10% alters the pore-wall structure to one characterized by interconnected rod-shaped TiC whiskers and a larger average pore size, which reduces the pore-wall strength and significantly decreases the compressive strength of the porous ceramic.

### 3.5. Thermal Conductivity

Figure 8a,b shows the thermal diffusivity and thermal conductivity of the as-received porous TiCO ceramic at room temperature and elevated temperatures, respectively. It can be seen from Figure 8a that porous TiCO ceramic with different catalyst content exhibits a low thermal diffusivity from 0.388 to 1.1 mm^2^/s at room temperature and a slight increment at elevated temperatures. As is well known, the heat conduction of the sample consists of three parts: solid heat transfer, gas heat transfer and radiation heat transfer. Generally, solid heat transfer could be decreased by the larger porosity and reduced solid contact area, and gas heat transfer can be lowered through controlling the pore size less than the average white path of gas molecules of 70 nm. The radiation heat transfer is proportional to the temperature. As the catalyst content increases from 0 to 5%, the porosity of the samples increases from 63% to 75%, while thermal conductivity decreases from 0.38 to 0.25 W/(m·K). This trend aligns with the conclusion drawn from numerous studies [6,7,14,15,21,35,37], which state that thermal conductivity is inversely proportional to the porosity of ceramics. When the catalyst content increases to 10%, the sample’s porosity further rises to 82%, and its thermal conductivity increases to 0.4 W/(m·K). This increase may be attributed to the formation of TiC whiskers, which create a solid-phase bridge across the pores, enhancing the solid heat transfer path. The thermal conductivities of the TiCO-0, TiCO-5, and TiCO-10 samples increase by 28%, 17%, and 18%, respectively, as the testing temperature rises from 25 °C to 600 °C. The primary cause of the increase in thermal conductivity with temperature is the enhancement of radiative heat transfer through the solid phase, which increases as the cube of the temperature [38]. The small, average nano-scale pore size of 52 nm in the TiCO-0 sample results in more solid-phase transfer paths, leading to a greater increase in thermal conductivity as the temperature rises. By adding the 5% and 10% catalyst, the reaction between pyrolytic carbon and TiO_2_ is enhanced, resulting in larger average nano-scale pore sizes of 110 nm and 139 nm, thereby reducing the solid-phase transfer path and diminishing the increase in thermal conductivity at elevated temperatures. Owing to the low density and thermal diffusivity, the TiCO-5 sample presents the lowest thermal conductivity among the three samples.

Figure 9 shows the schematic illustration of the structure of the porous TiCO ceramic. The increase in the catalyst content leads to an enhanced degree of carbothermal reduction. Therefore, the morphology of the pore wall changed from a honeycomb-like carbon structure composed of well-dispersed TiO particles to rod-shaped TiC whiskers interconnected with each other. The decrease in the gas heat transfer conduction path due to the average nano-scale pore size being lower than 70 nm, along with the reduction in the solid heat transfer path resulted from the interconnected rod-shaped TiC whiskers, both contributing to the lowest thermal conductivity of the TiCO-5 sample.

Figure 10 shows the comparison of thermal conductivity of the porous TiCO ceramic with other porous carbide ceramic reported in the literature [16,21,39,40]. As seen from Figure 10, the thermal conductivity at both room temperature and elevated temperature of the as-prepared porous TiCO ceramic is lower than that of porous ZrC and HfC ceramic. The porous high entropy (Zr_1/5_Hf_1/5_Nb_1/5_Ta_1/5_Ti_1/5_) ceramic fabricated by sintering method at 2000 °C in vacuum almost shows the same level of thermal conductivity of the as-prepared porous TiCO ceramic. These results demonstrate the feasibility and potential of the developed synthetic strategy in this study.

## 4. Conclusions

A novel, straightforward, one-step sintering process for fabricating porous TiCO ceramic with low thermal conductivity, high porosity, and robust compressive strength has been introduced. This method hinges on the in situ creation of a TiCO ceramic through the formation of a porous carbon skeleton resulting from the pyrolysis of phenolic resin and the following carbothermal reduction between the pyrolytic carbon and TiO_2_, facilitated by a Ni(NO_3_)_2_·6H_2_O catalyst. As the Ni(NO_3_)_2_·6H_2_O content increases from 0 to 10%, the carbothermal reduction becomes more sufficient, altering the pore-wall structure from TiO particles dispersed in a porous carbon skeleton to rod-shaped TiC whiskers that are interconnected. The resulting porous TiCO ceramic exhibits a dual modal pore-size distribution, comprising micro-scale pores with average size of about 50 μm and nano-scale pores with average size of 52–138 nm as the Ni(NO_3_)_2_·6H_2_O content varies from 0% to 10%. By adjusting the Ni(NO_3_)_2_·6H_2_O content, the as-received porous TiCO ceramic exhibit a porosity of 63–82%, a density of 0.72–1.05 g/cm^3^, a compressive strength of 0.07–1.89 MPa, and a room temperature thermal conductivity of 0.25–0.40 W/(m·K). Increasing the Ni(NO_3_)_2_·6H_2_O catalyst content can effectively enhance the carbothermal reduction between pyrolytic carbon and TiO_2_. However, there is still a TiO phase existing in sample TiCO-10. A further increase in the catalyst concentration may enhance the production of pure porous TiC ceramics. However, an excessive catalyst concentration could promote the formation of a larger pore size and TiC whiskers in the pore wall, resulting in increased porosity and thermal conductivity, while compressive strength may decrease to below 1 MPa, adversely affecting the insulation material’s performance. Complete carbothermal reduction may be achievable by minimizing the TiO_2_ particle size, the addition of sintering aids, or increasing the reaction temperature, which requires further investigation.

## Figures and Tables

**Figure 1 nanomaterials-14-00515-f001:**
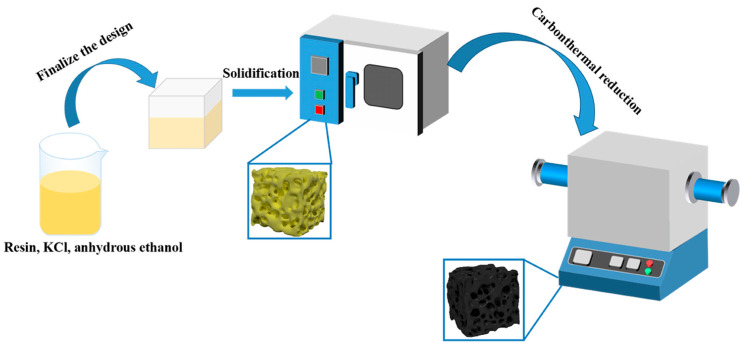
Fabrication process of the porous TiCO ceramic.

**Figure 2 nanomaterials-14-00515-f002:**
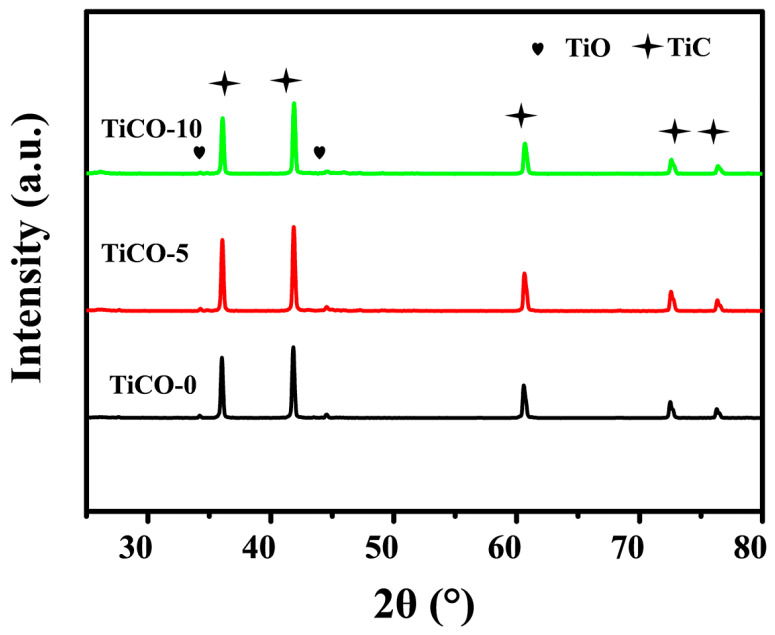
XRD patterns of the as-received TiCO ceramic with different catalyst contents.

**Figure 3 nanomaterials-14-00515-f003:**
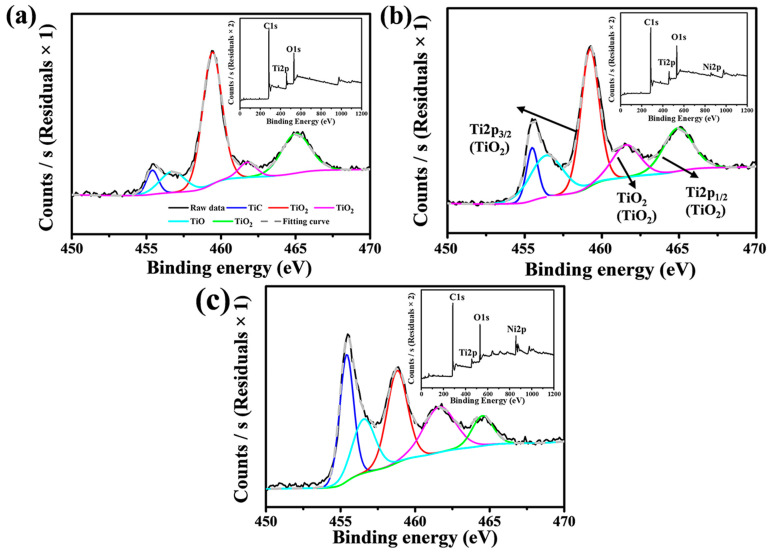
XPS patterns of the as-received TiCO ceramic: TiCO-0 (**a**), TiCO-5 (**b**), TiCO-10 (**c**).

**Figure 4 nanomaterials-14-00515-f004:**
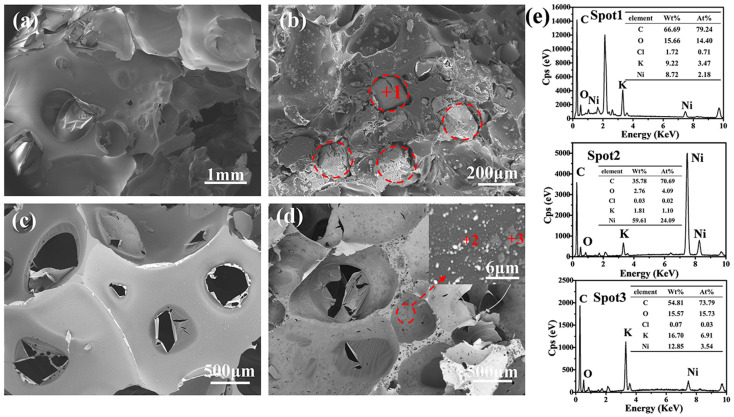
SEM images of the monolithic resin after cured (**a**) and sintered (**b**), and KCl/Ni(NO_3_)_2_·6H_2_O-5/resin after cured (**c**) and sintered (**d**), and the EDS results (**e**).

**Figure 5 nanomaterials-14-00515-f005:**
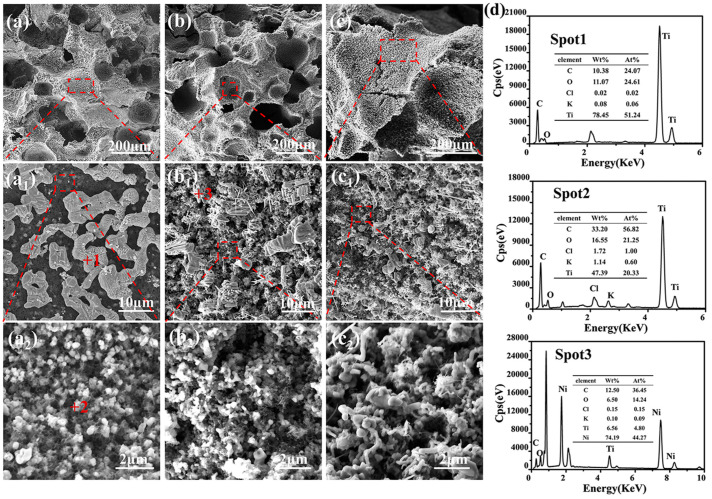
SEM images of the as-received porous TiCO ceramic: TiCO-0 (**a**), TiCO-5 (**b**), TiCO-10 (**c**), and the EDS results (**d**).

**Figure 6 nanomaterials-14-00515-f006:**
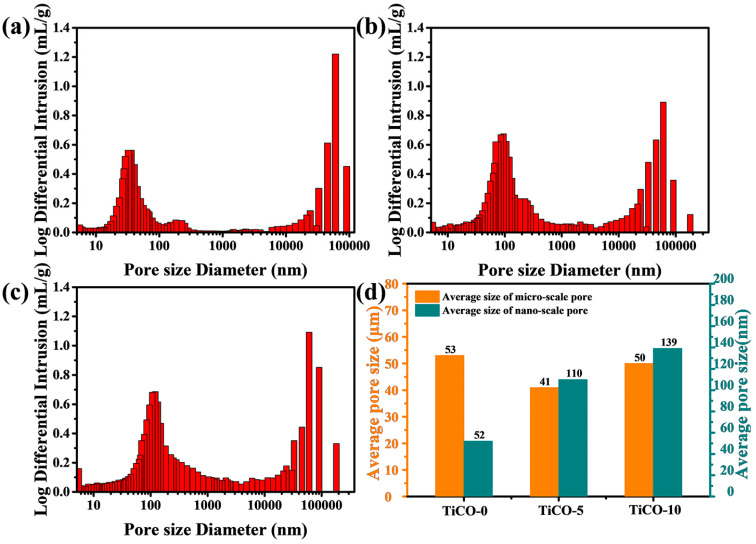
Log differential intrusion versus pore-size diameter of the as-received porous TiCO ceramic: TiCO-0 (**a**), TiCO-5 (**b**), TiCO-10 (**c**), average size of micro-scale pore and nano-scale pore (**d**) of the porous TiCO ceramic with different catalyst contents.

**Figure 7 nanomaterials-14-00515-f007:**
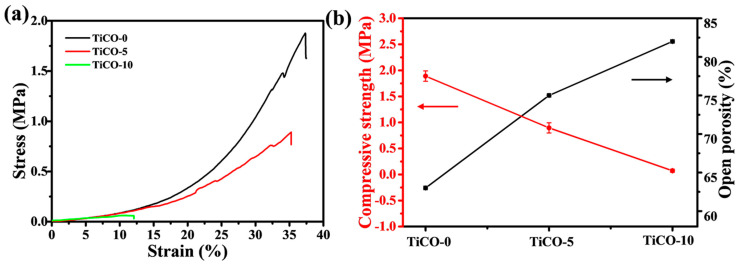
The strain–stress curves (**a**); Compressive strength and porosity (**b**) of the samples with different catalyst contents.

**Figure 8 nanomaterials-14-00515-f008:**
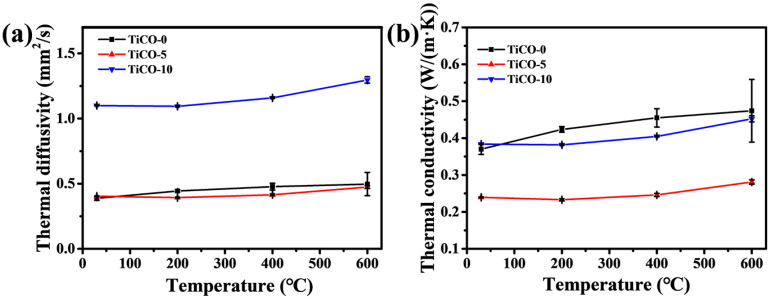
Thermal diffusivity (**a**) and thermal conductivity (**b**) of the samples with different catalyst contents.

**Figure 9 nanomaterials-14-00515-f009:**
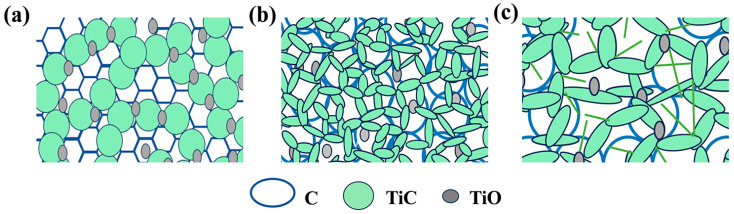
The schematic illustration of the pore wall of the porous TiCO ceramic: TiCO-0 (**a**), TiCO-5 (**b**), TiCO-10 (**c**).

**Figure 10 nanomaterials-14-00515-f010:**
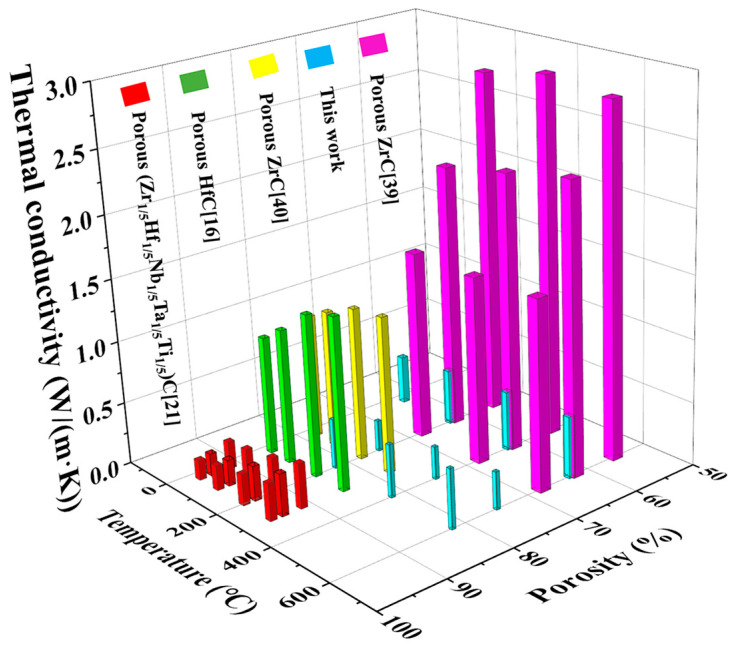
Comparison of the porosity and thermal conductivities of the porous TiCO ceramic to those of other porous carbide ceramics previously reported [16,21,39,40].

**Table 1 nanomaterials-14-00515-t001:** The open porosity, bulk density, compressive strength and thermal conductivity of the samples.

Sample	Open Porosity (%)	Bulk Density (g·cm^−3^)	Compressive Strength (MPa)	Thermal Conductivity W/(m·K)
TiCO-0	63	1.05	1.89 ± 0.100	0.38 ± 0.014
TiCO-5	75	0.87	0.89 ± 0.098	0.25 ± 0.001
TiCO-10	82	0.72	0.07 ± 0.030	0.40 ± 0.001

## Data Availability

The data that support the findings of this study are available from the corresponding author upon reasonable request.

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
