# Peer review of "Microstructure, Mechanical Property and Thermal Conductivity of Porous TiCO Ceramic Fabricated by In Situ Carbothermal Reduction of Phenolic Resin and Titania"

_nanomaterials, 2024, doi:10.3390/nano14060515_

Round 1

Reviewer 1 Report

Comments and Suggestions for Authors

This manuscript reports an experimental characterization of porous TiCO ceramic systems manufactured by a one-step sintering method, utilizing phenolic resin, TiO2 powder and KCl foaming agent as raw materials. The influence of varying catalyst content on the microstructure compressive strength and thermal conductivity of the final porous material was presented and analysed along with structural data by Xray diffraction and photoelectron and SEM microscopy.

Relevant insights of the microstructure and the pores formation were given also in relation to the compressive and thenal properties of the resulting materials.

The following issue should be addressed by the author before publication.

1)      Compression test data report strain % data but it is not clear how these values were measured or computed.

2)      The compressive curves of stress vs strain reported in Figure 7 are not commented on and discussed within the manuscript. The authors are invited to rationalise the curve trends with a specific focus on the ultimate failure strains%.

3)      The comments for the compressive data in relation to the open porosity % are not sufficiently clear.

4)      The thermal conductivity at different catalytic content (refers to figure 8) shows a very different path. The comments do not fully rationalise this behaviour thus the authors should emphasise more the possible effect of the catalyse content in light of the reported data.

 I would be happy to review the revised manuscript before publication. 

Comments on the Quality of English Language

Minore corrections are required for English and editing. 

Reviewer 2 Report

Comments and Suggestions for Authors

The authors need the revision of the manuscript for publication in nanomaterials journal. Some questions and suggestions are as follows;

1. If the Ni(NO3)2·6H2O catalyst content is further increased, we suggest that the discussion on porosity, mechanical strength, and thermal conductivity results will be added to the text. We believe that the determination of the optimized ratio of this system is very important.

2. We suggest that the authors provide specific information such as the physical properties of the phenolic resin powder. We believe that the authors should provide specific information related to phenolic resin to increase the understanding of this manuscript by journal readers.

3. The resolution of the picture in this manuscript is too low. In particular, the resolution of the inlet images and tables in Figures 3, 4, and 5 is too low, making it difficult for readers to understand this manuscript.

4. We suggest that authors should correct the typographic errors such as subscript issues in the whole manuscript.

5. The form of references described in the References part does not match the guidelines of the “nanomaterials” journal. The authors should revise the references’ form accurately.

Comments on the Quality of English Language

Minor editing of English language required. 

Round 2

Reviewer 2 Report

Comments and Suggestions for Authors

-